# Skew-Reflected-Gompertz Information Quantifiers with Application to Sea Surface Temperature Records

**Javier E. Contreras-Reyes [1],*** , **Mohsen Maleki [2]** and **Daniel Devia Cortés [3]**

[1]   Departamento de Estadística, Facultad de Ciencias, Universidad del Bío-Bío, Concepción 4081112, Chile
[2]   Department of Statistics, College of Sciences, Shiraz University, Shiraz 71946 85115, Iran;
     m.maleki.stat@gmail.com
[3]   Departamento de Evaluación de Pesquerías, Instituto de Fomento Pesquero, Valparaíso 2361827, Chile;
     ddeviac@gmail.com
*   Correspondence: jcontreras@ubiobio.cl or jecontrr@uc.cl; Tel.: +56-41-311-1199

**Abstract:** The Skew-Reflected-Gompertz (SRG) distribution, introduced by Hosseinzadeh et al. (J. Comput. Appl. Math. (2019) 349, 132–141), produces two-piece asymmetric behavior of the Gompertz (GZ) distribution, which extends the positive to a whole dominion by an extra parameter. The SRG distribution also permits a better fit than its well-known classical competitors, namely the skew-normal and epsilon-skew-normal distributions, for data with a high presence of skewness. In this paper, we study information quantifiers such as Shannon and Rényi entropies, and Kullback–Leibler divergence in terms of exact expressions of GZ information measures. We find the asymptotic test useful to compare two SRG-distributed samples. Finally, as a real-world data example, we apply these results to South Pacific sea surface temperature records.

**Keywords:** Skew-Reflected-Gompertz distribution; Gompertz distribution; entropy; Kullback–Leibler divergence; sea surface temperature

## 1. Introduction

The Skew-Reflected-Gompertz (SRG) distribution was recently introduced by [1] and corresponds to an extension of the Gompertz distribution [2], named after Benjamin Gompertz (1779–1865). It extends the positive dominion $\mathbb{R}_+$ to the whole of $\mathbb{R}$ by an extra parameter, $\varepsilon$, $-1 < \varepsilon < 1$, and produces two-piece asymmetric behavior of Gompertz (GZ) density. The SRG distribution has as particular cases the Reflected-GZ and GZ distributions, when $\varepsilon \to 1$ and $\varepsilon \to -1$, respectively. The SRG distribution family can also represent a suitable competitor against the skew-normal (SN, [3]) and epsilon-skew-normal (ESN, [4]) distributions as a way to fit asymmetrical datasets. Indeed, refs. [5,6] dealt with the frequentist and Bayesian inferences of ESN distribution. Contributions by [1] provided probability density function (pdf), cumulative distribution function (cdf), quantile function, moment-generating function (MGF), stochastic representation, the Expectation-Maximization (EM) algorithm for SRG parameter estimates and the Fisher information matrix (FIM).

Moreover, several recent investigations confirmed the usefulness of entropic quantifiers in the study of asymmetric distributions [3,7,8] and their applications to topics such as thermal wake [9], marine fish biology [3,8], sea surface temperature (SST), relative humidity measured in the Atlantic Ocean [10], and more. We build on the study of [3], which developed hypothesis testing for normality, i.e., if the shape parameter is close to zero. They considered the Kullback–Leibler (KL) divergence in terms of moments and cumulants of the modified SN distribution. Posteriorly, we consider a real-world data set of the anchovy condition factor for testing the shape parameter to decide if a food deficit produced by environmental conditions such as El Niño exists [11].

This work arose from a motivation to tackle the problem of determining the adequate pdf of SST [9,10]. Indeed, probabilistic modelling of SST is key for accurate predictions [9]. Therefore, we propose that the SRG model based on two-piece distributions could be more suitable for interpreting annual bimodal and asymmetric SST data. We also considered the existent results of Shannon and Rényi entropies, and KL divergence for GZ distributions for developed entropic quantifiers for SRG distributions. Posteriorly, we considered SST along the South Pacific and Chilean coasts from 2012 to 2014 to illustrate our results. Specifically, we introduced hypothesis testing developed by [12] for the SRG distribution, which is useful to compare two data sets with bimodal and asymmetric behavior such as SST.

## 2. The Skew-Reflected-Gompertz Distribution

The Gompertz (GZ, [2]) distribution is a continuous probability distribution with the following pdf

$$f(x|\sigma,\eta) = \frac{\eta}{\sigma}e^{\frac{x}{\sigma}}e^{-\eta(e^{\frac{x}{\sigma}}-1)}, \quad x \geq 0, \tag{1}$$

where $\sigma > 0$ and $\eta > 0$ are the scale and shape parameters, respectively, and are denoted by $X \sim GZ(\sigma,\eta)$. The mean and variance of $X$ are

$$\begin{aligned}
E(X) &= \sigma e^{\eta} Ei(-\eta), \\
Var(X) &= \sigma^2 e^{\eta} \tau,
\end{aligned} \tag{2}$$

respectively; where $Ei(z) = \int_{-z}^{\infty}\frac{e^{-u}}{u}du$, $\tau = -2\eta F(-\eta) + \gamma^2 + \frac{\pi^2}{6} + 2\gamma\log\eta + (\log\eta)^2 - e^{\eta}[Ei(-\eta)]^2$, $\gamma = 0.5772156649$ is the Euler constant and

$$F(z) = \sum_{k=0}^{+\infty}\frac{z^k}{k!(k+1)^3}.$$

The SRG distribution is an extension of the GZ proposed by [1]. If $Y$ follows, the SRG distribution is denoted by $Y \sim SRG(\mu,\sigma,\eta,\varepsilon)$ and has pdf

$$g(y|\mu,\sigma,\eta,\varepsilon) = \begin{cases} \frac{1}{2}f\left(\frac{\mu-y}{1+\varepsilon}\Big|\sigma,\eta\right), & y \leq \mu, \\ \frac{1}{2}f\left(\frac{y-\mu}{1-\varepsilon}\Big|\sigma,\eta\right), & y > \mu, \end{cases} \tag{3}$$

where $\mu \in \mathbb{R}$ is the location parameter and $\varepsilon \in (-1,1)$ is the slant parameter. Note that SRG is the GZ distribution when $\mu = 0$ and $\varepsilon \to -1$, GZ distribution with negative support when $\varepsilon \to 1$, and Reflected-GZ distribution when $\varepsilon = 0$. Also, the Reflected-GZ distribution corresponds to a particular case of a more general class of two-piece asymmetric distributions proposed by [13,14]. The mean, variance and MGF of $Y$ are

$$\begin{aligned}
E(Y) &= \mu - 2\varepsilon\sigma e^{\eta}Ei(-\eta), \\
Var(Y) &= \sigma^2\{\tau e^{\eta} + 2(1-\varepsilon^2)e^{2\eta}[Ei(-\eta)]^2\}, \\
M_Y(t) &= \frac{1}{2}\eta e^{\eta+\mu t}[(1-\varepsilon)F_{-\sigma t}(\eta) + (1+\varepsilon)F_{\sigma t}(\eta)],
\end{aligned} \tag{4}$$

respectively; where $F_s(z) = \int_1^{\infty}v^{s+1}e^{-vz}dv$. Jafari et al. [15] provide the MGF of $X$ using expansion series. However, (4) is considered a clearer expression that depends only on integral $F_s(z)$. See Section 4.1 for some details of the MLE EM-based algorithm related to SRG parameters.

According to [1], the SRG distribution can be re-parametrized in terms of GZ and Reflected-GZ distributions as

$$g(y|\mu,\sigma_+,\sigma_-,\eta) = p_1 f(\mu-y|\sigma_+,\eta)I_{(-\infty,\mu]}(y) + p_2 f(y-\mu|\sigma_-,\eta)I_{(\mu,+\infty)}(y), \tag{5}$$

where $\sigma_\pm = \sigma(1 \pm \varepsilon)$, $p_1 + p_2 = 1$, and $p_1 = \sigma_+/(\sigma_+ + \sigma_-) = (1+\varepsilon)/2$. Let $\mathbf{Y} = (Y_1, \ldots, Y_n)^\top$ be an i.i.d sample from the SRG distribution with parameters $(\mu, \sigma_\pm, \eta)$ and latent vectors $\mathbf{Z} = (\mathbf{Z}_1, \ldots, \mathbf{Z}_n)$, thus (5) can be equivalently represented as $(-1)^j(Y_i - \mu)|Z_{ij} = 1 \sim GZ(\sigma_\pm, \eta)$, $i = 1, \ldots, n$, $j = 1, 2$, where $\mathbf{Z}_i = (Z_{i1}, Z_{i2})^\top \sim Mult(1, p_1, p_2)$ is a multinomial vector, $P(Z_{i1} = z_{i1}, Z_{i2} = z_{i2}) = p_1^{z_{i1}} p_2^{z_{i2}}$, $z_{ij} = \{0,1\}$, and $z_{i1} + z_{i2} = 1$. Given that $P(Z_{i1} = 1) = P(Z_{i1} = 1, Z_{ik} = 0; \forall j \neq k)$, the complete log-likelihood function is

$$
\begin{aligned}
\ell(\mu, \sigma_+, \sigma_-, \eta | \mathbf{Y}, \mathbf{Z}) &= -n\log(2\sigma) + n(\eta + \log\eta) \\
&\quad + \sum_{i=1}^{n} \left[ z_{i1}\left( \frac{\mu - y_i}{\sigma_+} - \eta e^{\frac{\mu - y_i}{\sigma_+}} \right) + z_{i2}\left( \frac{y_i - \mu}{\sigma_-} - \eta e^{\frac{y_i - \mu}{\sigma_-}} \right) \right].
\end{aligned}
\tag{6}
$$

Conditional expectations of latent variables $\mathbf{Z}_i$ are given by

$$
\widehat{z}_{i1} = E[Z_{i1} | \widehat{\mu}, \widehat{\sigma}_+, \widehat{\sigma}_-, y_i] = \widehat{p}_1 \frac{f(\widehat{\mu} - y_i | \widehat{\sigma}_+, \widehat{\eta})}{g(y_i | \widehat{\mu}, \widehat{\sigma}_+, \widehat{\sigma}_-, \widehat{\eta})} I_{(-\infty, \widehat{\mu}]}(y_i),
\tag{7}
$$

$$
\widehat{z}_{i2} = 1 - \widehat{z}_{i1}, \quad i = 1, \ldots, n.
\tag{8}
$$

The E- and M-steps on the $(k+1)$th iteration of the EM algorithm are

**E-step.** From (6)–(8), we have

$$
\begin{aligned}
Q(\mu, \sigma_+, \sigma_-, \eta | \mu^{(k)}, \sigma_+^{(k)}, \sigma_-^{(k)}, \eta^{(k)}) &= E[\ell(\mu, \sigma_+, \sigma_-, \eta | \mathbf{Y}, \mathbf{Z}) | \mu^{(k)}, \sigma_+^{(k)}, \sigma_-^{(k)}, \eta^{(k)}] \\
&= -n\log(2\sigma) + n(\eta + \log\eta) \\
&\quad + \sum_{i=1}^{n} \left[ \widehat{z}_{i1}^{(k)}\left( \frac{\mu - y_i}{\sigma_+} - \eta e^{\frac{\mu - y_i}{\sigma_+}} \right) + \widehat{z}_{i2}^{(k)}\left( \frac{y_i - \mu}{\sigma_-} - \eta e^{\frac{y_i - \mu}{\sigma_-}} \right) \right].
\end{aligned}
$$

and

**M-step.** Update $\sigma_\pm$, by solving the following equation

$$
\sum_{i=1}^{n} \widehat{z}_{ij}^{(k)} \left( \eta^{(k)} \frac{|y_i - \mu^{(k)}|}{\sigma_\pm^2} e^{\frac{|y_i - \mu^{(k)}|}{\sigma_\pm}} - \frac{|y_i - \mu^{(k)}|}{\sigma_\pm^2} \right) = \frac{n}{2\sigma}.
$$

Update $\mu$ by solving the following equation

$$
\widehat{\mu}^{(k+1)} = \operatorname{argmax}_\mu \sum_{i=1}^{n} \left\{ \widehat{z}_{i1}^{(k)} \left( \frac{\mu - y_i}{\widehat{\sigma}_+^{(k+1)}} - \eta e^{\frac{\mu - y_i}{\widehat{\sigma}_+^{(k+1)}}} \right) + \widehat{z}_{i2}^{(k)} \left( \frac{\mu - y_i}{\widehat{\sigma}_-^{(k+1)}} - \eta e^{\frac{\mu - y_i}{\widehat{\sigma}_-^{(k+1)}}} \right) \right\}.
$$

Update $\eta$ by

$$
\widehat{\eta} = n \left( \sum_{i=1}^{n} \left\{ \widehat{z}_{i1}^{(k)} e^{\frac{\mu - y_i}{\widehat{\sigma}_+^{(k+1)}}} + \widehat{z}_{i2}^{(k)} e^{\frac{\mu - y_i}{\widehat{\sigma}_-^{(k+1)}}} \right\} \right)^{-1}.
$$

The EM-algorithm must be iterated until the sufficient convergence rule is satisfied:

$$
\| (\widehat{\mu}^{(k+1)}, \widehat{\sigma}_+^{(k+1)}, \widehat{\sigma}_-^{(k+1)}, \widehat{\eta}^{(k+1)}) - (\widehat{\mu}^{(k)}, \widehat{\sigma}_+^{(k)}, \widehat{\sigma}_-^{(k)}, \widehat{\eta}^{(k)}) \| < \tau,
$$

for a tolerance $\tau$ close to zero. The FIM for standard deviations of MLEs $(\widehat{\mu}, \widehat{\sigma}, \widehat{\eta}, \widehat{\varepsilon})$ and additional details of the EM-algorithm are described in [1].

## 3. Entropic Quantifiers

In the next section, we present the main results of entropic quantifiers for SRG distribution.

*3.1. Shannon Entropy*

The Shannon entropy (SE), introduced by [16] in the context of univariate continuous distributions, quantifies the information contained in a random variable $X$ with pdf $f(x)$ through the expression

$$H(X) = -\int_{-\infty}^{+\infty} f(x) \log f(x) dx. \tag{9}$$

The SE concept is attributed to the uncertainty of the information presented in $X$ [17]. Propositions 1 and 2 present the SE for GZ and SRG distributions, respectively.

**Proposition 1.** *[15]. The SE of $X \sim GZ(\sigma, \eta)$ is*

$$H(X) = \log\left\{\frac{B(1,1)}{\eta}\right\} - \sigma\eta - \frac{E(X)}{\sigma} + \sigma\eta M_X(\sigma^{-1}),$$

*where $B(\cdot, \cdot)$ is the usual Beta function and $E(X)$ is given in (2).*

Substituting $\mu = 0$ and $\varepsilon = -1$ into (4) (i.e., reducing SRG to its special case GZ), we obtain $M_X(\sigma^{-1}) = \eta e^\eta F_{-1}(\eta) = 1$. Therefore, $H(X)$ in Proposition 1 is reduced to

$$H(X) = -\log \eta - e^\eta E_i(-\eta), \tag{10}$$

i.e., the SE of the GZ random variable only depends on shape parameter $\eta$.

**Proposition 2.** *The SE of $Y \sim SRG(\mu, \sigma, \eta, \varepsilon)$ is*

$$H(Y) = \frac{1+\varepsilon}{2}\left\{H(X_{+\varepsilon}) - \log\left(\frac{1+\varepsilon}{2}\right)\right\} + \frac{1-\varepsilon}{2}\left\{H(X_{-\varepsilon}) - \log\left(\frac{1-\varepsilon}{2}\right)\right\},$$

*where $X_{\pm\varepsilon} \sim GZ(\sigma(1\pm\varepsilon), \eta)$ and $H(X_{\pm\varepsilon})$ are obtained using Proposition 1.*

**Proof.** From (3) and (9), we obtained

$$
\begin{aligned}
H(Y) &= -\int_{-\infty}^{+\infty} g(y|\mu, \sigma, \eta, \varepsilon) \log g(y|\mu, \sigma, \eta, \varepsilon) dy \\
&= -\frac{1}{2}\int_0^{+\infty} f\left(\frac{x}{1+\varepsilon}\Big|\sigma, \eta\right) \log\left\{\frac{1}{2}f\left(\frac{x}{1+\varepsilon}\Big|\sigma, \eta\right)\right\} dx \\
&\quad -\frac{1}{2}\int_0^{+\infty} f\left(\frac{x}{1-\varepsilon}\Big|\sigma, \eta\right) \log\left\{\frac{1}{2}f\left(\frac{x}{1-\varepsilon}\Big|\sigma, \eta\right)\right\} dx \\
&= -\frac{1}{2}\int_0^{+\infty} (1+\varepsilon)f(x|\sigma(1+\varepsilon), \eta) \log\left\{\frac{1+\varepsilon}{2}f(x|\sigma(1+\varepsilon), \eta)\right\} dx \\
&\quad -\frac{1}{2}\int_0^{+\infty} (1-\varepsilon)f(x|\sigma(1-\varepsilon), \eta) \log\left\{\frac{1-\varepsilon}{2}f(x|\sigma(1-\varepsilon), \eta)\right\} dx,
\end{aligned}
$$

which concludes the proof. □

From (10), given that $H(X_{\pm\varepsilon})$ only depends on shape parameter $\eta$, we obtain $H(X_{\pm\varepsilon}) = H(X)$, and $H(Y)$ only depends on $\eta$ and $\varepsilon$ parameters. Therefore,

$$H(Y) = -\log \eta - e^\eta E_i(-\eta) - \frac{1+\varepsilon}{2}\log\left(\frac{1+\varepsilon}{2}\right) - \frac{1-\varepsilon}{2}\log\left(\frac{1-\varepsilon}{2}\right). \tag{11}$$

Figure 1 illustrates SE behavior for random variable $Y$. We observed that SE increases when $\eta$ decreases. For each $\eta$, SE is maximized and minimized at $\varepsilon = 0$ (Reflected-GZ) and $\varepsilon \to -1$

(Truncated-GZ and GZ), respectively. More details appear in [3,8] for the SE expressions of other asymmetric distributions.

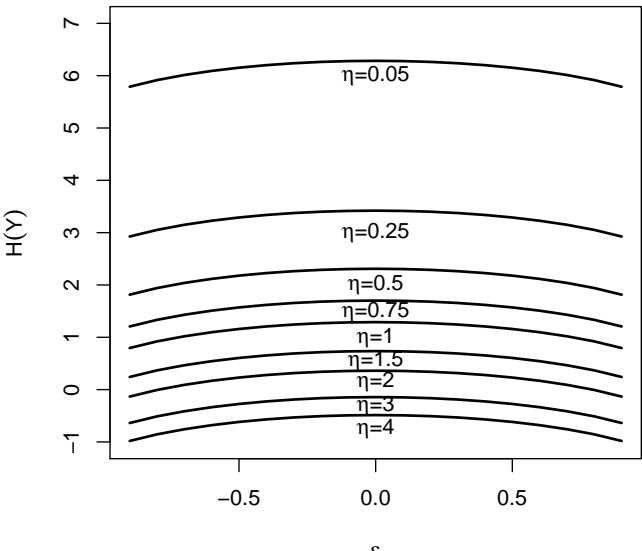

**Figure 1.** Shannon entropy of Skew-Reflected-Gompertz (SRG) distributions for $\varepsilon \in (-1, 1)$ and several values of $\eta$.

### 3.2. Rényi Entropy

The $\alpha$th-order Rényi entropy (RE), introduced by [18] in the context of univariate continuous distributions, extends the concept of SE information contained in a random variable $X$ with pdf $f(x)$ through a level $\alpha$, $\alpha \in \mathbb{N}$, $\alpha > 0$, and the expression

$$R_\alpha(X) = \frac{1}{1-\alpha} \log \int_{-\infty}^{+\infty} [f(x)]^\alpha dx. \tag{12}$$

RE information can be negative and is ordered with respect to $\alpha$, i.e., $R_{\alpha_1}(X) \geq R_{\alpha_2}(X)$ for any $\alpha_1 < \alpha_2$ (see, e.g., [7] and other properties of RE). From (12), the SE is obtained by the limit of $H(X) = \lim_{\alpha \to 1} R_\alpha(X)$ by applying l'Hôpital's rule to $R_\alpha(X)$ with respect to $\alpha$ (see e.g., [7]). The RE of the GZ and SRG distributions is presented in Propositions 3 and 4, respectively.

**Proposition 3.** *[15,19]. The RE of $X \sim GZ(\sigma, \eta)$ with $\alpha > 1$, $\alpha \in \mathbb{N}$, is*

$$R_\alpha(X) = -\frac{\log \alpha}{1-\alpha} + \log \frac{\eta}{\sigma} + \frac{1}{1-\alpha} \log \left\{ \sum_{j=0}^{\alpha-1} \binom{\alpha-1}{j} \frac{\Gamma(j+1)}{(\alpha\eta)^j} \right\},$$

*where $\Gamma(u) = \int_0^\infty t^{u-1} e^{-t} dt$ is the gamma function.*

**Proposition 4.** *The RE of $Y \sim SRG(\eta, \varepsilon)$ with $\alpha > 1$, $\alpha \in \mathbb{N}$, is*

$$R_\alpha(Y) = \frac{1}{1-\alpha} \log \left\{ \left(\frac{1+\varepsilon}{2}\right)^\alpha e^{(1-\alpha)R_\alpha(X_{+\varepsilon})} + \left(\frac{1-\varepsilon}{2}\right)^\alpha e^{(1-\alpha)R_\alpha(X_{-\varepsilon})} \right\},$$

*where $X_{\pm\varepsilon} \sim GZ(\sigma(1 \pm \varepsilon), \eta)$ and $R_\alpha(X_{\pm\varepsilon})$ are obtained using Proposition 3.*

**Proof.** From (3) and (12), we obtained

$$
\begin{aligned}
R_\alpha(Y) &= \frac{1}{1-\alpha} \log \int_{-\infty}^{+\infty} [g(y|\mu,\sigma,\eta,\varepsilon)]^\alpha dy, \\
&= \frac{1}{1-\alpha} \log \left\{ \int_0^{+\infty} \left[ \frac{1}{2} f\left( \frac{x}{1+\varepsilon} \Big| \sigma,\eta \right) \right]^\alpha dx + \int_0^{+\infty} \left[ \frac{1}{2} f\left( \frac{x}{1-\varepsilon} \Big| \sigma,\eta \right) \right]^\alpha dx \right\}, \\
&= \frac{1}{1-\alpha} \log \left\{ \left( \frac{1+\varepsilon}{2} \right)^\alpha \int_0^{+\infty} [f(x|\sigma(1+\varepsilon),\eta)]^\alpha dx + \left( \frac{1-\varepsilon}{2} \right)^\alpha \int_0^{+\infty} [f(x|\sigma(1-\varepsilon),\eta)]^\alpha dx \right\},
\end{aligned}
$$

which concludes the proof. $\square$

Figure 2a illustrates the behavior of RE for random variable $Y$ when $\alpha = 2$ (quadratic RE). As in the SE case, we also observed that RE increases when $\eta$ decreases and reaches maximum and minimum at $\varepsilon = 0$ (Reflected-GZ) and $\varepsilon \to -1$ (Truncated-GZ and GZ), respectively. When $\alpha = 5$ (or $\alpha > 2$) (see Figure 2b), RE decays faster than in the quadratic RE case as $\varepsilon \to -1$. More details appear in [7] for the RE expressions of other asymmetric distributions.

a)  b)

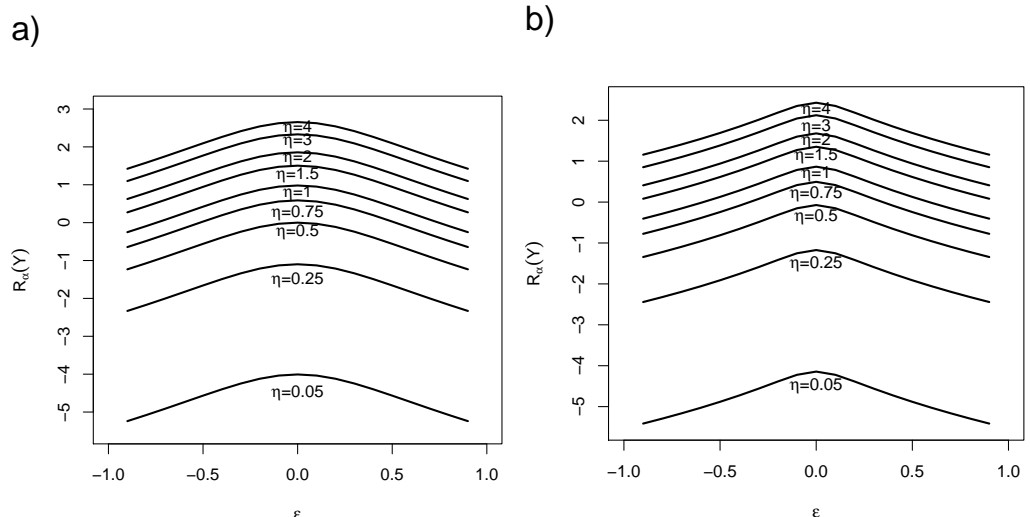

**Figure 2.** Rényi entropy of SRG distributions for $\sigma = 1$, $-1 < \varepsilon < 1$, several values of $\eta$ and (**a**) $\alpha = 2$ and (**b**) $\alpha = 5$ values.

*3.3. Kullback–Leibler Divergence*

The Kullback–Leibler (KL) divergence introduced by [20] in the context of univariate continuous distributions, extends the concept of SE between two random variables $X_1$ and $X_2$ with pdfs $f_1(x_1)$ and $f_2(x_2)$, respectively, through the expression

$$
K(X_1, X_2) = \int_{-\infty}^{+\infty} f_1(x) \log \left\{ \frac{f_1(x)}{f_2(x)} \right\} dx. \tag{13}
$$

The KL divergence measures the disparity between the pdfs of $X_1$ and $X_2$, and is non-negative, non-symmetric and zero only if $X_1 = X_2$ in distribution. Also, the KL divergence does not satisfy the triangular inequality (see, e.g., [8,17] for other properties of KL and other divergences). The KL divergence for two GZ and two SRG distributions are presented in Propositions 5 and 6.

**Proposition 5.** *[21]. The KL divergence between $X_1 \sim GZ(\sigma_1,\eta_1)$ and $X_2 \sim GZ(\sigma_2,\eta_2)$ is*

$$
K(X_1, X_2) = \log \left\{ \frac{e^{\eta_1}\sigma_2\eta_1}{e^{\eta_2}\sigma_1\eta_2} \right\} + e^{\eta_1} \left[ \left( \frac{\sigma_1}{\sigma_2} - 1 \right) E_i(-\eta_1) + \frac{\eta_2}{\eta_1^{\sigma_1/\sigma_2}} \Gamma\left( \frac{\sigma_1}{\sigma_2} - 1, \eta_1 \right) \right] - (\eta_1 + 1),
$$

where $\Gamma(u, v) = \int_v^\infty t^{u-1} e^{-t} dt$ is the upper incomplete gamma function.

**Proposition 6.** *The KL divergence between* $Y_1 \sim SRG(0, \sigma_1, \eta_1, \varepsilon_1)$ *and* $Y_2 \sim SRG(0, \sigma_2, \eta_2, \varepsilon_2)$ *is*

$$K(Y_1, Y_2) = \frac{1 + \varepsilon_1}{2} \left[ \log \left\{ \frac{1 + \varepsilon_1}{1 + \varepsilon_2} \right\} + K(X_{+\varepsilon_1}, X_{+\varepsilon_2}) \right] + \frac{1 - \varepsilon_1}{2} \left[ \log \left\{ \frac{1 - \varepsilon_1}{1 - \varepsilon_2} \right\} + K(X_{-\varepsilon_1}, X_{-\varepsilon_2}) \right],$$

*where* $X_{\pm \varepsilon_i} \sim GZ(\sigma_i(1 \pm \varepsilon_i), \eta_i)$, $i = 1, 2$, *and* $K(X_{\pm \varepsilon_1}, X_{\pm \varepsilon_2})$ *are obtained using Proposition 5.*

**Proof.** From (3) and (13), we obtained

$$
\begin{aligned}
K(Y_1, Y_2) &= \int_{-\infty}^{+\infty} g(x|0, \sigma_1, \eta_1, \varepsilon_1) \log \left\{ \frac{g(x|0, \sigma_1, \eta_1, \varepsilon_1)}{g(x|0, \sigma_2, \eta_2, \varepsilon_2)} \right\} dx, \\
&= \frac{1}{2} \int_0^{+\infty} f\left( \frac{x}{1 + \varepsilon_1} \Big| \sigma_1, \eta_1 \right) \log \left\{ \frac{f\left( \frac{x}{1+\varepsilon_1} \Big| \sigma_1, \eta_1 \right)}{f\left( \frac{x}{1+\varepsilon_2} \Big| \sigma_2, \eta_2 \right)} \right\} dx \\
&\quad + \frac{1}{2} \int_0^{+\infty} f\left( \frac{x}{1 - \varepsilon_1} \Big| \sigma_1, \eta_1 \right) \log \left\{ \frac{f\left( \frac{x}{1-\varepsilon_1} \Big| \sigma_1, \eta_1 \right)}{f\left( \frac{x}{1-\varepsilon_2} \Big| \sigma_2, \eta_2 \right)} \right\} dx, \\
&= \frac{1 + \varepsilon_1}{2} \left[ \log \left\{ \frac{1 + \varepsilon_1}{1 + \varepsilon_2} \right\} + \int_0^{+\infty} f(x|\sigma_1(1 + \varepsilon_1), \eta_1) \log \left\{ \frac{f(x|\sigma_1(1 + \varepsilon_1), \eta_1)}{f(x|\sigma_2(1 + \varepsilon_2), \eta_2)} \right\} dx \right] \\
&\quad + \frac{1 - \varepsilon_1}{2} \left[ \log \left\{ \frac{1 - \varepsilon_1}{1 - \varepsilon_2} \right\} + \int_0^{+\infty} f(x|\sigma_1(1 - \varepsilon_1), \eta_1) \log \left\{ \frac{f(x|\sigma_1(1 - \varepsilon_1), \eta_1)}{f(x|\sigma_2(1 - \varepsilon_2), \eta_2)} \right\} dx \right],
\end{aligned}
$$

which concludes the proof. $\quad\square$

More details appear in [3,8] for the KL divergence expressions of other asymmetric distributions. Using Proposition 6, the asymptotic KL divergence between $Y \sim SRG(0, \sigma, \eta, \varepsilon)$ and $X \sim GZ(\sigma, \eta)$ is

$$K(Y, X) \approx \frac{1 + \varepsilon}{2} \left[ \lim_{\varepsilon_2 \to -1} \log \left( \frac{1 + \varepsilon}{1 + \varepsilon_2} \right) + K(X_{+\varepsilon}, X) \right] + \frac{1 - \varepsilon}{2} \left[ \log \left( \frac{1 - \varepsilon}{2} \right) + K(X_{-\varepsilon}, X) \right],$$

as $\varepsilon_2 \to -1$. However, we see that $\log \left( \frac{1+\varepsilon}{1+\varepsilon_2} \right) = +\infty$ as $\varepsilon_2 \to -1$ and $K(Y, X)$ is not finite. However, from Proposition 6 the asymptotic KL divergence between $Y_1$ and $Y_2$ is

$$K(Y_1, Y_2) \approx K(X, Y) = \log \left( \frac{2}{1 - \varepsilon} \right) + K(X, X_{-\varepsilon}), \tag{14}$$

as $\varepsilon_1 \to -1$, where $X_{-\varepsilon} \sim GZ(\sigma(1 - \varepsilon), \eta)$. Therefore, while $K(Y, X)$ is not finite, $K(X, Y)$ is finite and can be used to study the disparity of $\varepsilon$ from $-1$. Thus, hypothesis testing for $H_0 : \varepsilon = -1$ can be addressed. Besides, we further study hypothesis testing for scale and shape parameters between two SRG distributions in Section 3.4. From (14), we also took that $K(Y_1, Y_2) \approx K(X, X_1)$ as $\varepsilon \to -1$, with $X_1 \sim GZ(2\sigma, \eta)$.

Figure 3 illustrates the KL divergence between two SRG distributions. We observed that for the critical points of $(\varepsilon_1, \varepsilon_2) \to \{(-1, 1); (1, -1)\}$, the KL divergence reaches the highest values and is close to zero in the other values [panels (a) and (b)]. For large $\eta$'s [panel (c)], the KL divergence is zero for a concentrated region of the dominion where $\varepsilon_1 = \varepsilon_2$.

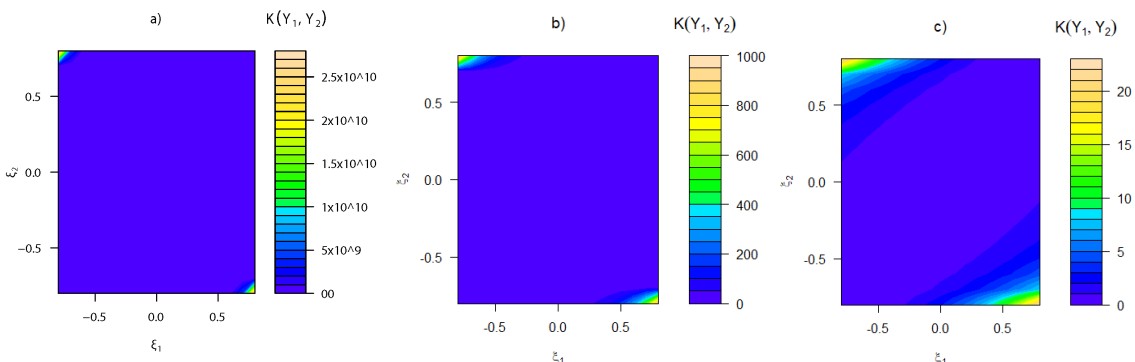

**Figure 3.** Plots of Kullback–Leibler (KL) divergence between $Y_1 \sim \mathrm{SRG}(0, \sigma_1, \eta_1, \varepsilon_1)$ and $Y_2 \sim \mathrm{SRG}(0, \sigma_2, \eta_2, \varepsilon_2)$ for values $\sigma_1 = \sigma_2 = 1$ and (**a**) $\eta_1 = \eta_2 = 0.25$; (**b**) $\eta_1 = \eta_2 = 3$; and (**c**) $\eta_1 = \eta_2 = 10$.

All information quantifiers and the EM algorithm for SRG distribution were implemented in [22].

### 3.4. Asymptotic Test

Consider two independent samples of sizes $n_1$ and $n_2$ from $Y_1$ and $Y_2$, respectively; where $\boldsymbol{\theta}$, $\boldsymbol{\theta}' \in \Theta \subset \mathbb{R}^p$, and $X_1$ and $X_2$ have pdfs $g(y; \boldsymbol{\theta}_1)$ and $g(y; \boldsymbol{\theta}_2)$, respectively; with $\boldsymbol{\theta}_i = (\sigma_i, \eta_i, \varepsilon_i)$, $i = 1, 2$. Suppose partition $\boldsymbol{\theta}_i = (\boldsymbol{\theta}_{i1}, \boldsymbol{\theta}_{i2})$, and assume $\boldsymbol{\theta}_{21} = \boldsymbol{\theta}_{11} \in \Theta_1 \subset \mathbb{R}^r$, so that $\boldsymbol{\theta}_{i2} \in \Theta \cap \Theta_1^c \subset \mathbb{R}^{p-r}$. Let $\widehat{\boldsymbol{\theta}}_i = (\widehat{\boldsymbol{\theta}}_{11}, \widehat{\boldsymbol{\theta}}_{i2})$ be the MLE of $\boldsymbol{\theta}_i = (\boldsymbol{\theta}_{11}, \boldsymbol{\theta}_{i2})$ for $i = 1, 2$, which corresponds to the MLE of the full model parameters $(\boldsymbol{\theta}_1, \boldsymbol{\theta}_2)$ under the null hypothesis $H_0 : \boldsymbol{\theta}_{21} = \boldsymbol{\theta}_{11}$. Thus, part b) of Corollary 1 in [12] establishes that if the null hypothesis $H_0 : \boldsymbol{\theta}_{22} = \boldsymbol{\theta}_{12}$ holds and $\frac{n_1}{n_1 + n_2} \xrightarrow[n_1, n_2 \to \infty]{} \lambda$, with $0 < \lambda < 1$, then

$$K_0 = \frac{2n_1 n_2}{n_1 + n_2} K(\widehat{\boldsymbol{\theta}}_1, \widehat{\boldsymbol{\theta}}_2) \xrightarrow[n_1, n_2 \to \infty]{d} \chi^2_{p-r}, \tag{15}$$

where $r = 3$ is the number of parameters of the SRG distribution (location parameter is not considered for KL divergence). Thus, a test of level $\alpha$ for the above homogeneity null hypothesis consists of rejecting $H_0$ if $K_0 > \chi^2_{p-r,1-\alpha}$, where $\chi^2_{p-r,\alpha}$ is the $\alpha$th percentile of the $\chi^2_{p-r}$-distribution.

As [3] stated, the proposed asymptotic test is only valid for regular conditions of the SRG distribution, in particular for a non-singular FIM. Therefore, given that the SRG distributions' FIM is singular at $\varepsilon \to \pm 1$ [1], the SRG model does not serve for testing the null hypothesis using (15) when $\varepsilon$ is close to $-1$ or $1$.

## 4. Application

### 4.1. Sea Surface Temperature Data

The spatial information and SST data analyzed in this study were recorded by a scientific observer (whose labor concerns biological sampling of fishes, incidental captures of birds, turtles and marine mammals. Biological sampling was complemented with information such as time, longline and hook features, number of buoys, baits, etc.) (SO) in the Chilean longline fleet (industrial and artisanal), which was oriented to capture swordfish (*Xiphias gladius*, [23]) from 2012 to 2014 (obtaining a sampling of 83% in 2012, 55% in 2013, 90% in 2014, and 75% in 2012–2014). The covered area of the study was at 21°31′–36°39′ LS and 71°08′–85°52′ LW (see Figure 4).

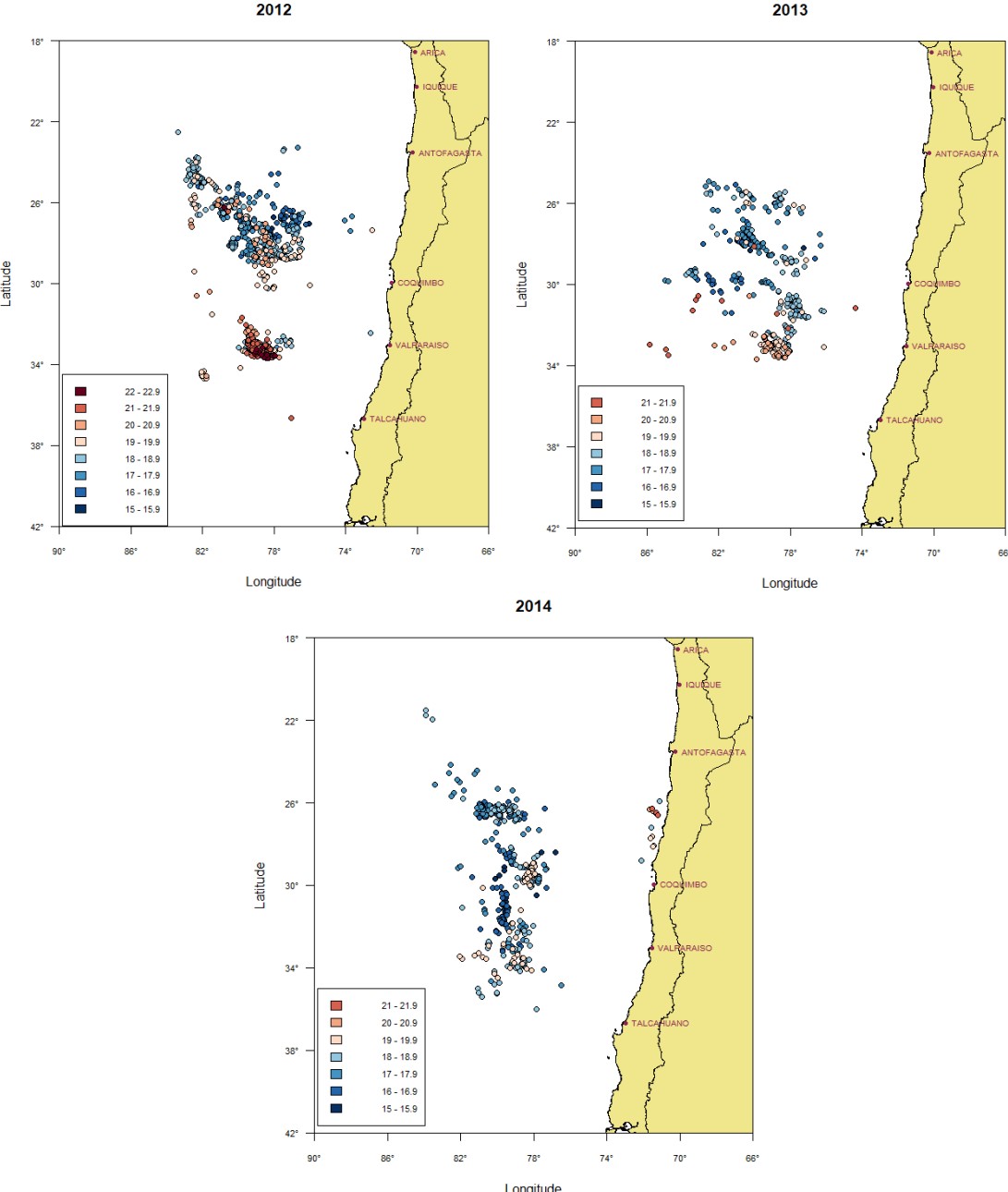

**Figure 4.** Spatial distribution of Sea Surface Temperature (SST) observations by year (21°31′–36°39′ LS, 71°08′–85°52′ LW).

SST records in swordfish captures are crucial for distributional analysis and fish abundance. Specifically, variations in SST are physical factors that control productivity, growth and migration of species [24]. In addition, SST is strongly correlated with atmospheric pressure at sea level and thus climatic time scales. Therefore, changes in SST overlap with ecosystem changes [25]. However, SST influence on ecosystems is not clear because other physical processes such as superficial warming, horizontal advection of currents, upwelling, etc. [11], modify SST. Therefore, SST anomalies could be symptomatic rather than causal.

#### 4.1.1. SRG Parameter Estimates

Considering the smallest Akaike (AIC) and Schwarz (BIC) information criteria, we observed in Table 1 that SRG performs better than the SN and ESN models (see Appendices A and B, respectively). In addition, Table 1 shows the estimated parameters (based on the EM algorithm presented in Section 2) for SST datasets by year assuming SRG distribution. In 2012, a negative $\varepsilon$ estimate corresponds to asymmetry to the right, and in 2013 and 2014 negative $\varepsilon$ and $\eta$ close to zero produce a two-piece distribution to fit "cold" and "warm" temperatures (Figure 5).

**Table 1.** Parameter estimates and their respective standard deviations (SD) for SST by year based on SRG, epsilon-skew-normal (ESN) and skew-normal (SN) models. For each model, log-likelihood function $\ell(\theta)$, $\theta = (\mu, \sigma, \eta, \varepsilon)$, Akaike's (AIC) and Bayesian (BIC) information criteria, and goodness-of-fit tests (Kolmogorov–Smirnov (K–S), Anderson–Darling (A–D), and Cramer–von Mises, (C–V)) are also reported with respective p-values in parentheses.

| Year | Model | Param. | Estim. | (S.D) | $\ell(\theta)$ | AIC | BIC | K–S | A–D | C–V |
|---|---|---|---|---|---|---|---|---|---|---|
| 2012 ($n = 774$) | SRG | $\mu$ | 17.992 | 0.103 | $-1401.896$ | 2811.793 | 2830.399 | 0.044 (0.095) | 2.014 (0.090) | 0.214 (0.242) |
| | | $\sigma$ | 2.590 | 0.067 | | | | | | |
| | | $\eta$ | 1.444 | 0.027 | | | | | | |
| | | $\varepsilon$ | $-0.207$ | 0.075 | | | | | | |
| | ESN | $\theta$ | 18.000 | 0.031 | $-1507.534$ | 3021.069 | 3035.023 | 0.118 ($<0.01$) | 26.417 ($<0.01$) | 2.059 ($<0.01$) |
| | | $\omega$ | 1.657 | 0.015 | | | | | | |
| | | $\epsilon$ | $-0.418$ | 0.069 | | | | | | |
| | SN | $\xi$ | 16.777 | 0.114 | $-1404.581$ | 2815.161 | 2829.116 | 0.041 (0.143) | 1.752 (0.126) | 0.198 (0.271) |
| | | $\omega$ | 5.199 | 0.043 | | | | | | |
| | | $\lambda$ | 2.527 | 0.311 | | | | | | |
| 2013 ($n = 415$) | SRG | $\mu$ | 17.935 | 0.061 | $-687.420$ | 1382.839 | 1398.942 | 0.082 (0.010) | 2.632 (0.042) | 0.491 (0.041) |
| | | $\sigma$ | 1.112 | 0.026 | | | | | | |
| | | $\eta$ | 0.432 | 0.021 | | | | | | |
| | | $\varepsilon$ | $-0.108$ | 0.029 | | | | | | |
| | ESN | $\theta$ | 17.600 | 0.046 | $-716.375$ | 1438.750 | 1450.827 | 0.089 ($<0.01$) | 7.721 ($<0.01$) | 0.970 (0.002) |
| | | $\omega$ | 1.328 | 0.026 | | | | | | |
| | | $\epsilon$ | $-0.376$ | 0.092 | | | | | | |
| | SN | $\xi$ | 16.598 | 0.200 | $-691.531$ | 1389.063 | 1401.140 | 0.066 (0.054) | 2.002 (0.092) | 0.328 (0.113) |
| | | $\omega$ | 3.812 | 0.054 | | | | | | |
| | | $\lambda$ | 2.421 | 0.617 | | | | | | |
| 2014 ($n = 439$) | SRG | $\mu$ | 17.454 | 0.048 | $-653.082$ | 1314.164 | 1330.502 | 0.092 ($<0.01$) | 2.848 (0.033) | 0.533 (0.032) |
| | | $\sigma$ | 0.896 | 0.020 | | | | | | |
| | | $\eta$ | 0.375 | 0.020 | | | | | | |
| | | $\varepsilon$ | $-0.106$ | 0.025 | | | | | | |
| | ESN | $\theta$ | 17.200 | 0.053 | $-703.748$ | 1413.496 | 1425.750 | 0.109 ($<0.01$) | 11.996 ($<0.01$) | 1.529 ($<0.01$) |
| | | $\omega$ | 0.956 | 0.035 | | | | | | |
| | | $\epsilon$ | $-0.384$ | 0.090 | | | | | | |
| | SN | $\xi$ | 16.146 | 0.098 | $-666.984$ | 1339.968 | 1352.222 | 0.096 ($<0.01$) | 4.055 ($<0.01$) | 0.711 (0.011) |
| | | $\omega$ | 3.245 | 0.045 | | | | | | |
| | | $\lambda$ | 3.434 | 0.618 | | | | | | |

To evaluate the goodness-of-fit test, the Kolmogorov–Smirnov (K–S), Anderson–Darling (A–D), and Cramer–von Mises (C–V) tests were considered for all models, commonly used to analyze the goodness-of-fit test of a particular distribution see, e.g., [26]). Considering a 95% confidence level, SRG fits perform well for 2012 and 2013, and on a 90% confidence level, the SRG fit performs well for 2014.

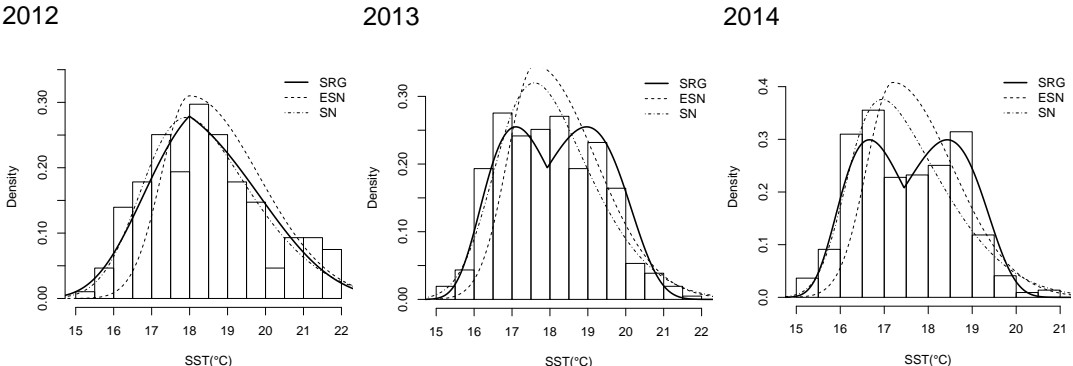

**Figure 5.** MLE fit of SRG, ESN and SN models for SST data by year.

### 4.1.2. Information Quantifiers and Asymptotic Test

Parameters estimated from the SRG model and presented in Table 1 are used to perform the quantifiers of Sections 3.1–3.3 for SST in each year and for the asymptotic test of Section 3.4 for comparing SST between two years. The results of these analyses are shown in Table 2. In Table 2, $K_0 = \widehat{K}(Y_1, Y_2)$ represents the KL divergence between the years $Y_1$ (column) and $Y_2$ (row).

The first quantifiers (SE and RE) illustrate that the highest information of SST is obtained by SE and increases with the increment of years. For all RE, the highest information of SST is obtained in 2012 and is negative for 2013 and 2014 and similar during that period. Differences in information between SE and RE are produced by the independency of SE with parameter $\sigma$, while RE depends on three parameters as in Proposition 4.

In addition, the asymptotic test presented in Table 2 is analogous for all the years in both groups. In fact, the null hypothesis $H_0 : \boldsymbol{\theta}_1 = \boldsymbol{\theta}_2$ is rejected at a 95% confidence level. This rejection is reinforced by high values of statistics $K_0$, produced by a high sample size of both groups ($n_1$ and $n_2$).

**Table 2.** SRG Shannon, $H(Y)$, and Rényi, $R_\alpha(Y)$, $\alpha = 2, 3, 4$, entropies for SST data. For each year, the KL divergence $K_0 = \widehat{K}(Y_1, Y_2)$, statistic and its respective *p*-values of Equation (15) are reported. All reported $K_0$ estimates considered the estimated parameters and sample size *n* in Table 1.

| Year | Quantifier | 2012 | 2013 | 2014 |
|---|---|---|---|---|
| | $H(Y)$ | 0.765 | 0.781 | 2.754 |
| | $R_2(Y)$ | 0.384 | $-0.362$ | $-0.365$ |
| | $R_3(Y)$ | 0.252 | $-0.417$ | $-0.418$ |
| | $R_4(Y)$ | 0.163 | $-0.457$ | $-0.457$ |
| 2012 | $K_0$ | - | 0.266 | 0.911 |
| | Statistic | - | 143.740 | 520.41 |
| | *p*-value | - | <0.01 | <0.01 |
| 2013 | $K_0$ | 0.080 | - | 0.071 |
| | Statistic | 43.192 | - | 30.233 |
| | *p*-value | <0.01 | - | <0.01 |
| 2014 | $K_0$ | 0.143 | 0.043 | - |
| | Statistic | 80.327 | 18.282 | - |
| | *p*-value | <0.01 | <0.01 | - |

## 5. Conclusions

We have presented a methodology to compute the Shannon and the Rényi entropy and the Kullback–Leibler divergence for the family of Skew-Reflected-Gompertz distributions. Our methods consider the information quantifiers previously computed for the Gompertz distribution. Explicit formulas for Shannon and Rényi entropies (in terms of the Gompertz, Shannon and

Rényi entropies, respectively), and the Kullback–Leibler divergence (using incomplete gamma function) facilitate easy computational implementation. Additionally, given the regularity conditions accomplished by the Skew-Reflected-Gompertz distribution, specifically by the Fisher information matrix convergence when $\varepsilon$ is in $(-1, 1)$, an asymptotic test for comparing two groups of datasets was developed.

The statistical application to South Pacific sea surface temperature was given. We first carried out SRG goodness-of-fit tests in samples over three years, where we find strong evidence (a 95% confidence level) for 2012, and moderate evidence (a 90% confidence level) for 2013 and 2014. The results show that the proposed methodology serves to compare two sets of samples, Skew-Reflected-Gompertz distributed. The proposed asymptotic test is therefore useful to detect anomalies in sea surface temperature, linked to extreme events influenced by environmental conditions [11,24,25]. We encourage researchers to consider the proposed methodology for further investigations related to environmental datasets [1].

**Author Contributions:** J.E.C.-R. and M.M. wrote the paper and contributed reagents/analysis/materials tools; J.E.C.-R. and D.D.C. conceived, designed and performed the experiments and analyzed the data. All authors have read and approved the final manuscript.

**Funding:** This research received no external funding.

**Acknowledgments:** We are grateful to the Instituto de Fomento Pesquero (IFOP) for providing access to the data used in this work. Special thanks to Fernando Espíndola for his helpful insights and discussion on an early version of this paper. The SST datasets and R codes used in this work are available upon request to the corresponding author. The authors thank the editor and two anonymous referees for their helpful comments and suggestions.

**Conflicts of Interest:** The authors declare that there is no conflict of interest in the publication of this paper.

## Abbreviations

The following abbreviations are used in this manuscript:

| | |
|---|---|
| A–D | Anderson–Darling |
| AIC | Akaike's information criterion |
| BIC | Bayesian information criterion |
| C–V | Cramer–von Mises |
| CDF | Cumulative distribution function |
| EM | Expectation maximization |
| ESN | Epsilon-skew-normal |
| FIM | Fisher information matrix |
| GZ | Gompertz |
| K–S | Kolmogorov–Smirnov |
| KL | Kullback–Leibler |
| MGF | Moment-generating function |
| MLE | Maximum Likelihood Estimator |
| PDF | Probability density function |
| RE | Rényi entropy |
| SD | Standard deviation |
| SE | Shannon entropy |
| SN | Skew-normal |
| SRG | Skew-Reflected-Gompertz |
| SST | Sea surface temperature |

## Appendix A. The Epsilon-Skew-Normal Distribution

The epsilon-skew-normal distribution [4,27] in its location-scale version is denoted as $\mathrm{ESN}(\theta, \varpi, \epsilon)$. It can be derived from a more general class of two-piece asymmetric distributions proposed by [14], by considering the standardized normal kernel $\phi(\cdot)$ (zero mean and variance 1), denoted as $N(0, 1)$,

as the density $f$ and the functions $a(\epsilon) = 1 + \epsilon$ and $b(\epsilon) = 1 - \epsilon$. If $Z \sim \text{ESN}(\theta, \varpi, \epsilon)$, thus $Z$ has pdf given by

$$h(z|\theta, \varpi, \epsilon) = \begin{cases} \phi\left(\frac{\theta - z}{\varpi(1+\epsilon)}\right), & z \leq \theta, \\ \phi\left(\frac{z - \theta}{\varpi(1-\epsilon)}\right), & z > \theta, \end{cases} \tag{A1}$$

where $Z = \theta + \varpi X$ for location $\theta \in \mathbb{R}$ and scale $\varpi > 0$ parameters. The mean and variance of $Z$ are

$$\begin{aligned} E(Z) &= \theta - 4\varpi\epsilon/\sqrt{2\pi}, \\ Var(Z) &= \frac{\varpi^2}{\pi}[(3\pi - 8)\epsilon^2 + \pi], \end{aligned}$$

and the MGF of $X$ is given by

$$M_X(t) = (1+\epsilon)e^{\frac{(1+\epsilon)^2 t^2}{2}}\Phi[-(1+\epsilon)t] + (1-\epsilon)e^{\frac{(1-\epsilon)^2 t^2}{2}}\Phi[(1-\epsilon)t],$$

where $\Phi(\cdot)$ is the cdf of standardized Gaussian distribution.

## Appendix B. The Skew-Normal Distribution

Let $X$ be a skew-normal (SN, [28]) random variable denoted as $X \sim \text{SN}(\xi, \omega, \lambda)$. The pdf of $X$ is given by

$$f(x; \lambda) = 2\phi(z)\Phi(\lambda z), \tag{A2}$$

with $z = (x - \xi)/\omega$. The SN model with the density (A2) is explained by its stochastic representation

$$X \stackrel{d}{=} \xi + \delta|U_0| + \sqrt{1 - \delta^2}U, \tag{A3}$$

where $\delta = \lambda/\sqrt{1 + \lambda^2}$, $X$ is represented as a linear combination of Gaussian $U$ and a half-Gaussian $|U_0|$ variable, and $U_0 \sim N(0, 1)$ and $U \sim N(0, \omega^2)$ are independent (Theorem 1 of [29]). From (A3), the mean and variance of $X$ are $E(X) = \xi + \sqrt{2/\pi}\delta$ and $Var(X) = \omega^2 - (2/\pi)\delta^2$, respectively.

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
