# Peer review of "Skew-Reflected-Gompertz Information Quantifiers with Application to Sea Surface Temperature Records"

_mathematics, doi:10.3390/math7050403_

Round 1
Reviewer 1 Report
See attached file.

Author Response
Dear Reviewer:
We would like to acknowledge this careful revision of our manuscript mathematics-477539: "Skew-Reflected-Gompertz information quantiers with application to sea surface temperature records". We are grateful that this manuscript can be considered for publication after major revision. We also thank the reviewer for all their valuable comments and constructive criticism. We have included (see attached file below), a detailed point-by-point response to all the reviewer's comments and suggestions. The comments from the reviewer are listed in cursive italic letters.
Best regards,
Dr. Javier E. Contreras-Reyes
Corresponding author

Reviewer 2 Report
Comments on "Skew-Reflected-Gompertz information quantifiers with application to sea surface temperature records", submitted to Mathematics.
By, Contreras-Reyes, Maleki & Cortez
The paper derives information theoretic measures for the SRG distribution and proposes a goodness-of-fit procedure.
My main questions relate to the goodness-of-fit procedure.
From the practical point of view:
- First of all, why should an SRG family be considered appropriate for the SST data?
- Is there any reason to believe the discontinuity (see between 17-18 deg C) to be realistic in an SST distribution? (e.g. see fig 5). In particular, are there other potential families of distributions for these data?- In the figure caption, it would be nice if the authors would remind us again, what the various acronyms stand for. In fact, the entire paper consists of far too many acronyms, making reading difficult.
- The plotted figures are not convincing enough arguments that the families of distributions considered in the paper are appropriate for the SST.
From both practical & theoretical point of view:
- The goodness of fit section (subsection 5.4) is much too short. Please elaborate on the details in this section.
- What are X1, X2, Y1, Y2? Of course you have used these notations earlier in the paper, but you should introduce these again in this section.
- Perhaps cite the theorem from [27] that is about the limit theorem (Eq 13).
- Please elaborate on the null & the alternative hypotheses. Why is one interested in doing this test? What is the physical interpretation for the SST data?
- The need for considering divergence measures etc should be discussed.
- The advantage of using divergence measure based tests for your data, over other tests should be explained.
- Have you tried fitting other families of distributions to these data? Can you show & comment on your findings?
- Advantage of your procedure over any existing one should be highlighted.
- Please mention source of the data in your acknowledgements & please give the web link, in case these are freely downloadable for research.
Finally, English language correction is recommended.
Author Response
Dear Reviewer:
We would like to acknowledge this careful revision of our manuscript mathematics-477539: "Skew-Reflected-Gompertz information quantifiers with application to sea surface temperature records". We are grateful that this manuscript can be considered for publication after major revision. We also thank the reviewer for all their valuable comments and constructive criticism. We have included (see attached file below), a detailed point-by-point response to all the reviewer's comments and suggestions. The comments from the reviewer are listed in cursive italic letters.
Best regards,
Dr. Javier E. Contreras-Reyes
Corresponding author

Round 2
Reviewer 1 Report
See attached file.

Author Response
Dear Reviewer:
We would like acknowledge this careful revision of our manuscript mathematics-477539:
"Skew-Reflected-Gompertz information quantiers with application to sea surface temperature
records". We are grateful that this manuscript can be considered for publication after mayor
revision. We also thank the reviewer for all their valuable comments and constructive criticism.
We have included (see attached file below), a detailed point-by-point response to all the
reviewer's comments and suggestions. The comments from the reviewer are listed in cursive
italic letters.

Reviewer 2 Report
No further comments on the revised version.
Author Response
Dear Reviewer:
We would like acknowledge this careful revision of our manuscript mathematics-477539:
"Skew-Reflected-Gompertz information quantiers with application to sea surface temperature
records". We also thank the reviewer for all their valuable comments and constructive criticism.
Round 3
Reviewer 1 Report
I have no further comments.